# Mosquito abundance and diversity in central Ohio, USA vary among stormwater wetlands, retention ponds, and detention ponds and their associated environmental parameters

James Radl[1], Luis Martínez Villegas[1], Joseph S. Smith[2], R. Andrew Tirpak[2], Kayla I. Perry[1], Deirdre Wetmore[3], Elena Tunis[1], Jack Smithberger[2], Henry Schuellerman[4], Dom Magistrado[1], Ryan J. Winston[2,4], Sarah M. Short[1]*

1 Department of Entomology, The Ohio State University, Columbus, Ohio, United States of America,
2 Department of Food, Agricultural, and Biological Engineering, The Ohio State University, Columbus, Ohio, United States of America, 3 School of Environment and Natural Resources, The Ohio State University, Columbus, Ohio, United States of America, 4 Department of Civil, Environmental, and Geodetic Engineering, The Ohio State University, Columbus, Ohio, United States of America

* short.343@osu.edu

**Data Availability Statement:** All relevant data are within the paper and its Supporting Information files. Sanger sequences have also been uploaded

## Abstract

Mosquitoes (Diptera: Culicidae) are one of the most impactful pests to human society, both as a nuisance and a potential vector of human and animal pathogens. Mosquito larvae develop in still aquatic environments. Eliminating these habitats near high human density or managing them to reduce the suitability for mosquitoes will reduce mosquito populations in these human environments and decrease the overall negative impact of mosquitoes on humans. One common source of standing water in urban and suburban environments is the water that pools in stormwater control measures. Previous studies have shown that some stormwater control measures generate large numbers of mosquitoes while others harbor none, and the reason for this difference remains unclear. Our study focuses on elucidating the factors that cause a stormwater control measure to be more or less suitable for mosquitoes. During the summers of 2021 and 2022, we collected and identified mosquito larvae from thirty stormwater control measures across central Ohio to assess variation in mosquito abundance and diversity among sites. Our goal was to determine if specific types of stormwater control measures (retention ponds, detention ponds, or constructed wetlands) harbored different abundances of mosquitoes or different community structures. We also assessed environmental parameters of these sites to elucidate their effects on mosquito abundance and diversity. Overall, we recorded the highest number of mosquito larvae and species in constructed wetlands. However, these sites were dominated by the innocuous species, *Culex territans*. Conversely, detention ponds held fewer mosquitoes but a higher proportion of known vector species, including *Culex pipiens* and *Aedes vexans*. The total number of mosquitoes across all sites was correlated with higher vegetation, more shade, lower water temperatures, and lower pH, suggesting stormwater control measures with these features may also be hotspots for mosquito proliferation.

to GenBank, and accession numbers are listed in
S3 File.

**Funding:** JR, LMV, ET, DM, and SMS were funded
by the Ohio State University Infectious Diseases
Institute and the Ohio State University College of
Food, Agricultural, and Environmental Sciences.
JSS, RAT, DW, JS, HS, and RJW were funded by
the Ohio State University College of Food,
Agricultural, and Environmental Sciences and the
Ohio State University College of Engineering. The
funders had no role in study design, data collection
and analysis, decision to publish, or preparation of
the manuscript.

**Competing interests:** The authors have declared
that no competing interests exist.

## Introduction

Mosquitoes (Diptera: Culicidae) are one of the most impactful pests to human society. They are a common nuisance pest in most parts of the world. The adult females of many species readily bite humans and can inhibit outdoor activities when populations are high [1]. Additionally, several species have the capacity to spread pathogens to humans, pets, and livestock if they are infected with pathogens when they bite their hosts. Throughout the world, mosquitoes transmit the human pathogens that cause malaria, Zika fever, dengue, Chikungunya, yellow fever, and many more leading to hundreds of millions of human disease cases and several hundred thousand deaths annually [2]. Although diseases like yellow fever and malaria have not been a widespread concern in the continental United States since the mid-20th century [3], autochthonous cases of malaria and dengue occasionally occur primarily in the southern US [4, 5]. Additionally, mosquitoes in the US can transmit lethal pathogens that cause West Nile fever, eastern and western equine encephalitis, St. Louis encephalitis, and diseases that affect pets and livestock, such as dog heartworm [6]. One of the primary ways to decrease the incidences of these diseases is by decreasing the overall populations of their mosquito vectors [3].

All mosquito larvae are aquatic and develop in shallow, still water that exists typically for a minimum of about seven days for them to reach adulthood [7, 8]. One of the most common and voluminous sources of standing water in urban or suburban environments is the permanent storage that is part of stormwater control measures (henceforth called SCMs) [9]. Worldwide, SCMs are constructed in cities to mitigate the negative effects of excessive stormwater [10–13]. SCMs have become increasingly common in the past 50 years, both in the United States and around the world [12–15]. Many SCMs retain stormwater for either a few days or permanently (depending on the type of SCM) to reduce flow velocity and promote treatment mechanisms that improve the quality and reduce the quantity of stormwater released to rivers or other bodies of water [7, 11, 12, 14]. In addition to their stormwater management benefits, SCMs may also be used to recharge groundwater, provide locations for human recreation, or support natural wildlife [16–18].

There are many parameters to consider in the design of SCMs, such as the storage capacity, surface area, cost, dewatering rate, plant species selection, treatment objectives, and the extent of available land [19–23]. These parameters are often the primary factors in determining which type of SCM will be built [7, 21, 22]. Retention ponds (also known as "wet ponds"), constructed wetlands, and detention ponds (also known as "dry ponds") are all common types of SCMs that retain standing water which may serve as anthropogenic habitats for mosquito larval development. Retention ponds and constructed wetlands are designed to permanently retain large volumes of water, whereas detention ponds are designed to drain the majority of temporarily captured water within a few days [7, 22]. In Ohio and some other jurisdictions, detention ponds typically feature a permanent micropool/forebay at the outlet and inlet of the system, [12, 21, 24] but often retain larger volumes for longer periods of time when their outlets are clogged because of a lack of maintenance [25–27].

Although SCMs have been known to harbor mosquito larvae [28–30], there is a lack of information about which types of SCMs are less attractive or provide a less suitable environment for mosquito development. Irwin et al. highlight that many SCMs foster no or very few mosquitoes all year while other SCMs generate large populations of mosquitoes throughout the year and over multiple years [9]. Many studies of mosquitoes in SCMs focus on constructed wetlands (e.g., [31–36]) because of their widespread adoption for stormwater control and because the public associates wetlands with mosquitoes [37]. However, other types of SCMs may be even more suitable for certain mosquitoes. For example, detention ponds, which frequently dry out and refill with runoff during rain events, create an ideal habitat for

floodwater mosquitoes like *Aedes vexans* (Meigen) [38]. The few studies that have compared mosquitoes in constructed wetlands and the more common detention and retention ponds found that constructed wetlands produce comparable or lower numbers of mosquitoes compared to other types of SCMs [28, 35].

While SCM type may be generally predictive of productive mosquito habitats, numerous specific biotic and abiotic parameters of individual SCM may also affect mosquito production. These include parameters associated with the water itself, such as temperature, presence of predators, presence of aquatic vegetation, availability of nutrients, and organic matter concentration [28, 30, 33, 35, 39–42]. Additionally, parameters of the surrounding environment may also play an important role in the abundance and diversity of mosquitoes, such as vegetation surrounding the SCMs, shading of the SCM by vegetation or structures, and watershed land use [32, 33, 43]. Some of these parameters are correlated with SCM types (e.g., constructed wetlands are planted with native plants and therefore typically have higher vegetation coverage than other SCM types) but may also vary among sites (e.g., retention and detention ponds are typically not planted but may have dense vegetation when overrun with invasive plants like *Typha* spp.). Because there is variation among sites, it is helpful to identify which abiotic and biotic parameters are most impactful on mosquito production independently from SCM type.

The goal of this research was to compare mosquito larval abundance and diversity among three different types of SCMs: retention ponds, detention ponds, and constructed wetlands. Additionally, we assessed the correlations between mosquito abundance and several environmental parameters of each SCM, including: water temperature, turbidity, conductivity, pH, shade, presence of vegetation within the water, and presence of vegetation on land adjacent to the water. Because different species of mosquitoes have varying impacts on humans (e.g., their affinity for biting humans and/or their potential for transmitting pathogens), we assessed the suitability of SCMs as mosquito larval development sites on a species-specific basis [33]. Findings from this study can be used to identify SCM parameters that foster mosquito populations as well as maintenance activities that may be required to limit the proliferation of mosquitoes in these systems.

## Material and methods

### Sampling locations

Thirty SCM sites across the greater Columbus, Ohio, USA metropolitan area were selected for surveillance in this project. These sites consisted of three distinct SCM types: retention ponds, detention ponds, and constructed wetlands (n = 3 SCM types; n = 10 sites/type). These sites had a variety of land managers (e.g., cities, schools, businesses), maintenance practices, drainage areas, storage volumes, age, surface areas, and surrounding environments. Candidate SCMs were selected by type, ease of access, safety of research staff, and willingness of each land manager to participate in the study and not by *a priori* predictions of their mosquito communities. Written consent to use each given site was obtained from managers of private properties and appropriate government offices for public lands. To protect the confidentiality of the site owners, each site was assigned a unique identifier in place of site names or locations.

### Sampling procedure

We sampled these sites every two weeks from the first week of June through the first week of October (n = 9 sampling periods/year) during the summers of 2021 and 2022. For each retention pond and constructed wetland, we identified five subsites for sampling which were equally spaced along the SCM circumference (n = 5 subsites/site for retention ponds and constructed wetlands). Because detention ponds typically only hold water in an inlet forebay and an outlet

**Table 1. List of parameters recorded for each sample.**

| Variable | Description | Values/Units |
|---|---|---|
| Sampling period | Biweekly period when samples were taken | 1–9 |
| SCM type | SCM design type | Retention pond (RP), detention pond (DP), or constructed wetland (CW) |
| Site | Individual SCM basin | Unique identifier for each SCM (n = 30) |
| Subsite | Sampling locations within each site | 1–5 for RP and CW; inlet or outlet for DP |
| Depth | Depth of water where sample was collected | 5 cm and 15 cm (or closest depth) |
| pH | pH of the water | |
| Conductivity | Conductivity of the water | µS/cm |
| Temperature | Temperature of the water | ˚C |
| Turbidity | Turbidity of the water | Ntu |
| Shade | Qualitative assessment of shade over subsite | 1 (direct sunlight), 2 (partial shade), 3 (full shade) |
| Vegetation within water | Qualitative assessment of vegetation in subsite | 1 (no vegetation), 2 (vegetation below water surface only), 3 (vegetation above water surface) |
| Vegetation on land | Qualitative assessment of vegetation on land adjacent to subsite | 1 (no vegetation), 2 (mowed grass only), 3 (unmowed grass or other vegetation < 1 m height), 4 (vegetation > 1 m height) |
| Total mosquitoes | Count of all mosquito larvae and pupae | |
| APS | Mosquito abundance per sample | |

Each sample consists of the number of larvae collected from 700ml water sample as well as measurements of the subsite's parameters.

micropool, we only chose a single subsite close to the inlet and outlet each (n = 2 subsites/site for detention ponds). At each subsite, we collected samples at 5 cm and 15 cm depth (n = 2 samples/subsite). If we were not able to locate these specific depths at each subsite during a given sampling period, we sampled at the next closest depth and separately recorded the depth. In cases where the subsite was completely dry or inaccessible, we did not sample the subsite during that period.

During each sampling period, we collected 700 ml samples of water at each depth/subsite/site using a standard 350 ml larval dipper (as described in [44]). The water was poured into a bin where we visually examined it for any mosquito larvae and pupae, regardless of size or instar. All larvae and pupae were transferred to specimen cups and brought back to the lab for identification. In addition to counting larvae, we recorded the turbidity of the water using a Hach 2100Q portable turbidimeter (Hach Company, Loveland, CO, USA) as well as pH, water temperature, and conductivity *in situ* using a YSI handheld multiparameter meter (models 1030, 1020, or 090; YSI inc. Yellow Springs, OH, USA) at each subsite. We also made a qualitative assessment of the shade provided by trees, shrubs, or other nearby features. Finally, we assessed the vegetation within the water and on land adjacent to the subsite. A list of variables recorded from each sample is provided in Table 1.

### Mosquito identification

Mosquito larvae were identified based on a list of morphological characteristics and keys in [45–48]. We allowed all pupae time to eclose, then identified them based on their adult characteristics mentioned in the previous materials. After identification, we counted and recorded all larvae and pupae of each species. Any larvae and pupae that died or were otherwise unidentifiable were recorded as unidentified and, therefore, we included them in our total mosquito counts but not in the counts of individual species. A complete dataset can be found in S1 File. We also classified mosquitoes as pest or innocuous species based on whether they are known to bite humans and potentially transmit pathogens to humans. To verify our morphological

identifications, we randomly selected at least four specimens (or all specimens if fewer than four) of each species for molecular barcoding using the cytochrome c oxidase I (COI) gene, as described by [49]. We limited our morphological identification of *Anopheles* spp. to the genus level and, therefore, this taxon was excluded from our COI verifications. Genomic DNA was extracted from individual whole larvae of the other taxa using ZymoBIOMICS DNA Miniprep kits (Zymo Research, Irvine CA, USA, catalog no. D4304) following the manufacturer's instructions. The COI gene was amplified in each sample using PCR and primers LCO1490 and HCO2198 [50]. The resulting PCR product was purified using Invitrogen PureLink Quick PCR Purification kits (Thermo Fisher Scientific Baltics UAB, Vilnius, Lithuania, catalog no. K310001) and submitted to the Genomic Shared Resource at the Ohio State University's Comprehensive Cancer Center for Sanger sequencing. Forward and reverse COI sequences were manually trimmed and merged, then queried in the National Center for Biotechnology Information GenBank using their Basic Local Alignment Search Tool [51, 52]. The highest percent identity was compared to our morphological identifications for verification.

## Statistical analyses

All statistical analyses were conducted in R version 4.1.1 [53]. A copy of the R code and output for all statistical analyses can be found in S2 File. Generalized linear mixed models (GLMM) were developed using the package 'glmmTMB' [54] to evaluate the effects of SCM type (retention ponds, detention ponds, constructed wetlands) and sampling period (1–9) as fixed effects on the mosquito abundance per sample (APS; total number of mosquitoes collected at a site divided by the number of samples taken). Each GLMM included site as a random effect, and separate models were developed for the data collected in 2021 and 2022. Because datasets from both years had a disproportionate number of datapoints with zero APS, we used a zero-altered two-part model (also known as a "hurdle" model) for our data [55]. A zero-altered GLMM (ZA-GLMM) produces two models: the first (binomial model) uses our covariates to model the presence (APS > 0) versus absence (APS = 0) of mosquitoes, and the second (conditional model) uses our covariates to model the positive count data. Because the positive data were overdispersed and left-skewed, we used a negative binomial distribution for all GLMMs. The ANOVA function in the 'car' package [56] was used to test for the significance of SCM type and sampling period as predictors in both the binomial and conditional models. If SCM type was significant, Tukey contrasts were calculated using the emmeans function in the 'emmeans' package [57].

Relationships among the mosquito species, total mosquito abundance, SCM types, and environmental variables (pH, temperature, conductivity, turbidity, vegetation in water, vegetation on land, and shade) were compared using partial least squares canonical analysis (PLSCA; [58]). Partial least squares methods are well-suited for datasets with a high number of predictor variables. This approach is also more appropriate when the predictor variables show a high level of multicollinearity because they are designed to maximize the covariances among variables [58]. PLSCA deflates the matrix of response variables based on the regression of the response variables, rather than the predictor variables, which allows for a more symmetrical role of both types of variables in determining the derived latent variables that form the model. PLSCA was performed using the 'mixOmics' package [59]. From our PLSCA, we generated correlation circle plots and similarity matrices (using the network function and represented as clustered image maps or CIMs) to visualize the correlations between pairs of variables. The correlation values of the similarity matrices are approximations of Pearson correlation coefficients and can be interpreted similarly (see [60] for discussion of how these values are calculated). Variables with correlations higher than 0.4 or lower than -0.4 were reported as

correlations of interest. We also ran indicator species analyses using the 'indicspecies' package [61] to identify species that are common among multiple sites of the same SCM type, affirming an ecological relationship between the SCM type and mosquito species.

We compared the alpha and beta diversity of mosquito communities among SCM types. We estimated the diversity of each site based on the total number of mosquitoes collected from the sites for each year (2021 and 2022 datasets were assessed separately). We evaluated alpha diversity by estimating species richness, species evenness, and the Shannon diversity index calculated using the 'vegan' package [62]. Because the variances were unequal among SCM types for all three metrics, we used nonparametric Kruskal-Wallis tests to determine if alpha diversity of mosquitoes differed among the three SCM types using the base R 'stats' package [53]. If SCM type was significant, post-hoc pairwise Wilcoxon tests were calculated using the Bonferroni correction for multiple comparisons. To assess differences in mosquito communities among SCM types, we created pairwise dissimilarity matrices using the Bray-Curtis method. Permutational multivariate analysis of variance (PERMANOVA) tests were used to determine if SCM types harbored different communities of mosquitoes. Differences in mosquito communities among SCM types were visualized using non-metric multidimensional scaling (NMDS). PERMANOVA and NMDS analyses were conducted using the 'vegan' package [62]. If SCM type was significant, pairwise PERMANOVA tests were conducted using the 'pairwiseAdonis' package [63].

## Results

### Mosquito species

We collected a total of 5761 larvae and pupae across 30 SCM sites in 2021 and 2022. Excluding the 46 pupae that died before eclosion, we identified 5715 larvae and pupae into nine mosquito species along with the *Anopheles* spp. group (Table 2). We chose to group all *Anopheles* into a single taxon because we were unable to consistently and accurately separate *Anopheles punctipennis* (Say), *Anopheles quadrimaculatus* Say, and *Anopheles perplexens* Ludlow based on the morphological characters provided in the keys. Overall, our samples were dominated by five

**Table 2. Abundance and pest status of mosquito species found in SCMs.**

| Species | Pest Status | 2021 Totals | | | 2022 Totals | | | Total |
| --- | --- | --- | --- | --- | --- | --- | --- | --- |
| | | DP | RP | CW | DP | RP | CW | |
| *Aedes vexans* | Pest | 190 | 4 | 231 | 63 | 0 | 94 | 582 |
| *Anopheles* spp. | Pest | 36 | 174 | 195 | 59 | 238 | 175 | 877 |
| *Culiseta inornata* | Pest | 0 | 0 | 0 | 0 | 0 | 14 | 14 |
| *Culex erraticus* | Pest | 19 | 164 | 51 | 8 | 499 | 56 | 797 |
| *Culex pipiens* | Pest | 160 | 0 | 219 | 611 | 11 | 127 | 1128 |
| *Culex restuans* | Pest | 1 | 0 | 10 | 23 | 95 | 102 | 231 |
| *Culex salinarius* | Pest | 0 | 0 | 24 | 4 | 0 | 152 | 180 |
| *Culex territans* | Innocuous | 32 | 365 | 879 | 19 | 43 | 499 | 1837 |
| *Psorophora ferox* | Pest | 0 | 0 | 0 | 0 | 0 | 2 | 2 |
| *Uranotaenia sapphirina* | Innocuous | 0 | 4 | 46 | 0 | 12 | 5 | 67 |
| Unidentified | NA | 9 | 2 | 18 | 7 | 4 | 6 | 46 |
| All species | | 447 | 713 | 1673 | 794 | 902 | 1232 | 5761 |

Samples were collected in the Columbus, Ohio metropolitan area in the summers of 2021 and 2022. The SCM types are listed as detention pond (DP), retention pond (RP), and constructed wetland (CW). Mosquitoes were classified as pest species if they are known to bite humans and potentially vector human pathogens, otherwise, they are classified as innocuous.

*Culex* species (*Culex territans* Walker, *Culex erraticus* (Dyar and Knab), *Culex pipiens* L., *Culex restuans* Theobald, and *Culex salinarius* Coquillett), which represented 73% of the total mosquitoes. *Anopheles* spp. made up 15% of our total, and the only species of *Aedes* (*Ae. vexans*) made up another 10%. We collected a small number of *Uranotaenia sapphirina* (Osten Sacken) (1%). We also collected a few individuals of *Psorophora ferox* (Humboldt) and *Culiseta inornata* (Williston) in a single sample each. The most common species in our samples was the innocuous species, *Cx. territans* (32%). Morphological identification was validated by COI sequencing, and all morphological identifications matched those obtained by Sanger sequencing (S3 File).

In both years, we collected the highest total abundance of mosquitoes from constructed wetlands, followed by retention ponds, and the fewest from detention ponds (Table 2). Additionally, during both years, constructed wetlands harbored the highest number of species across all sites, including all ten taxa in 2022. Because these abundance numbers are conflated with the number of samples we took from each SCM type, we also calculated the mosquito abundance per sample (APS) and found that either constructed wetlands (2021) or detention ponds (2022) harbored the highest APS (Table 3). In both years, retention ponds had the lowest APS (Table 3).

## Effects of SCM type on mosquito abundance

In 2021, we found that SCM type significantly affected the presence of mosquitoes (Fig 1A and Table 4). Mosquitoes were significantly more likely to be present in constructed wetlands compared to detention ponds and retention ponds, but no difference was observed between detention ponds and retention ponds (Fig 1A and Table 5). In sites positive for mosquitoes, SCM type did not significantly affect APS (Fig 1A and Table 4). In 2022, we did not find a significant effect of SCM type on the presence or the APS of mosquitoes (Fig 1B and Table 6). During both years, sampling period had a significant effect on the presence of mosquitoes, but only significantly impacted the APS of mosquitoes in 2022 (Fig 1 and Tables 4 and 6).

## Effects of environmental parameters on mosquito abundance

Using PLSCA, several abiotic and biotic environmental variables were associated with the abundance of mosquitoes (Fig 2; S1 Table). During both years, total mosquito abundance was

**Table 3. Mosquito abundance per sample (APS) found in SCMs.**

| Species | 2021 APS | | | 2022 APS | | | Mean |
|---|---|---|---|---|---|---|---|
| | DP | RP | CW | DP | RP | CW | |
| *Aedes vexans* | 0.73 | <0.01 | 0.29 | 0.19 | 0 | 0.11 | 0.14 |
| *Anopheles* spp. | 0.14 | 0.20 | 0.24 | 0.18 | 0.26 | 0.21 | 0.22 |
| *Culiseta inornata* | 0 | 0 | 0 | 0 | 0 | 0.02 | <0.01 |
| *Culex erraticus* | 0.07 | 0.18 | 0.06 | 0.02 | 0.55 | 0.07 | 0.20 |
| *Culex pipiens* | 0.62 | 0 | 0.27 | 1.9 | 0.01 | 0.15 | 0.28 |
| *Culex restuans* | <0.01 | 0 | 0.01 | 0.07 | 0.11 | 0.12 | 0.06 |
| *Culex salinarius* | 0 | 0 | 0.03 | 0.01 | 0 | 0.18 | 0.04 |
| *Culex territans* | 0.12 | 0.41 | 1.09 | 0.06 | 0.05 | 0.59 | 0.46 |
| *Psorophora ferox* | 0 | 0 | 0 | 0 | 0 | <0.01 | <0.01 |
| *Uranotaenia sapphirina* | 0 | <0.01 | 0.06 | 0 | 0.01 | 0.01 | 0.02 |
| Unidentified | 0.03 | <0.01 | 0.02 | 0.02 | <0.01 | <0.01 | 0.01 |
| All species | 1.72 | 0.80 | 2.07 | 2.42 | 1.00 | 1.46 | 1.43 |

Samples were collected in the Columbus, Ohio metropolitan area in the summers of 2021 and 2022. The SCM types are listed as detention pond (DP), retention pond (RP), and constructed wetland (CW).

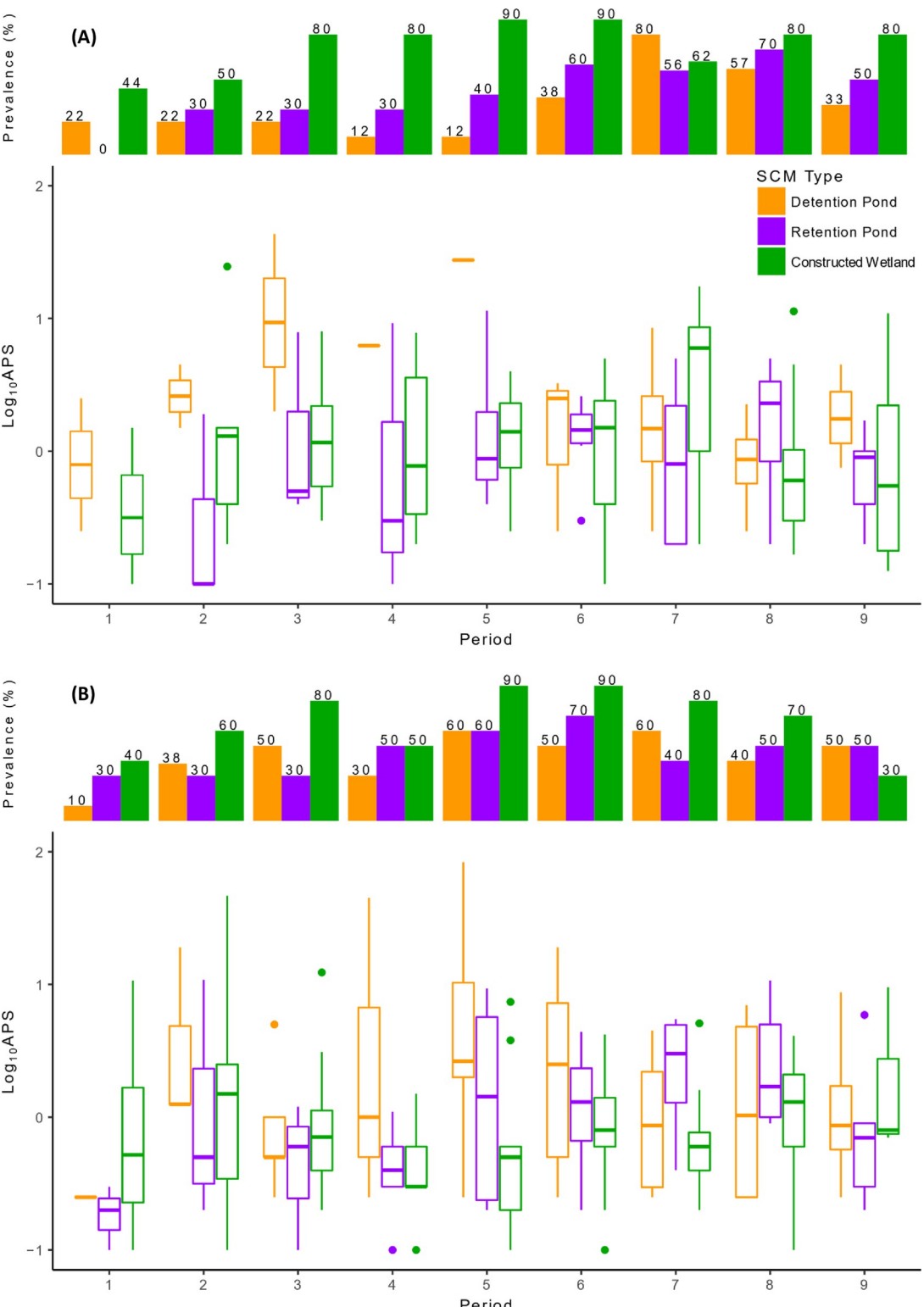

**Fig 1.** Prevalence and APS of mosquitoes found in SCM sites during (A) 2021 and (B) 2022. The prevalence of mosquitoes indicates the percentage of sites with at least one mosquito larva collected (positive count data). The APS (mosquito abundance per sample) was calculated by dividing the total number of mosquitoes collected in a site during a single sampling period by the number of samples taken. Sites with zero APS were removed. The box plots show the median, quartiles, minimum, maximum, and outliers among the sites belonging to each SCM type and period.

**Table 4. Results of the ZA-GLMM estimating mosquito presence (binomial model) and APS (conditional model) in 2021.**

| Model | Factor | $\chi^2$ | df | p-value |
|---|---|---|---|---|
| Binomial | SCM type | 10.56 | 2 | < 0.01* |
| | Sampling period | 25.02 | 8 | < 0.01* |
| Conditional | SCM type | 2.24 | 2 | 0.33 |
| | Sampling period | 6.04 | 8 | 0.64 |

*p-values were determined with ANOVAs comparing GLMMs with and without the selected factor; significant values are marked with an asterisk.

**Table 5. Results of the pairwise comparisons between SCM types in ZA-GLMMs estimating mosquito APS in 2021.**

| Contrast | estimate | se | df | t-ratio | p-value |
|---|---|---|---|---|---|
| DP-RP | 0.66 | 0.93 | 223 | 0.70 | 0.76 |
| DP-CW | 3.02 | 0.98 | 223 | 3.01 | < 0.01* |
| RP-CW | 2.37 | 0.94 | 223 | 2.52 | 0.03* |

*p-values were determined with a Tukey's test to determine if pairs of SCM types were significantly different; significant values are marked with an asterisk.

**Table 6. Results of the ZA-GLMM estimating mosquito presence (binomial model) and APS (conditional model) in 2022.**

| Model | Factor | $\chi^2$ | df | p-value |
|---|---|---|---|---|
| Binomial | SCM type | 4.16 | 2 | 0.12 |
| | Sampling period | 23.13 | 8 | < 0.01* |
| Conditional | SCM type | 4.33 | 2 | 0.11 |
| | Sampling period | 18.49 | 8 | 0.02* |

*p-values were determined with ANOVAs comparing GLMMs with and without the selected factor; significant values are marked with an asterisk.

positively correlated with vegetation on land, vegetation within water, shade, and constructed wetland SCMs and negatively correlated with temperature, pH, and retention pond SCMs. Retention ponds were also negatively correlated with *Cx. pipiens* in 2021 and 2022 and *Cx. territans* in 2021. Detention ponds were positively correlated with *Cx. pipiens* in 2022 and negatively correlated with *Cx. erraticus* in 2021 and 2022 and *Anopheles* spp. in 2021. *Cx. territans* was positively correlated with shade in 2021 and 2022, vegetation on land in 2021 and 2022, vegetation within water in 2021, and shade in 2021. *Anopheles* spp. was positively correlated with vegetation within water in 2021. Finally, *Cx. salinarius* was positively correlated with vegetation on land in 2022.

Indicator species analyses revealed that several species were indicative of constructed wetlands and retention ponds, but no species were identified as indicators of detention ponds during either year. *Ur. sapphirina* in 2021 (stat = 0.678, p = 0.03) and *Cx. salinarius* in 2022 (stat = 0.698, p = 0.025) were indicative of constructed wetlands. Additionally, *Cx. erraticus* (2021: stat = 0.857, p = 0.01; 2022: stat = 0.831, p = 0.015) and *Anopheles* spp. (2021: stat = 0.880, p = 0.03) were indicative of both constructed wetlands and retention ponds.

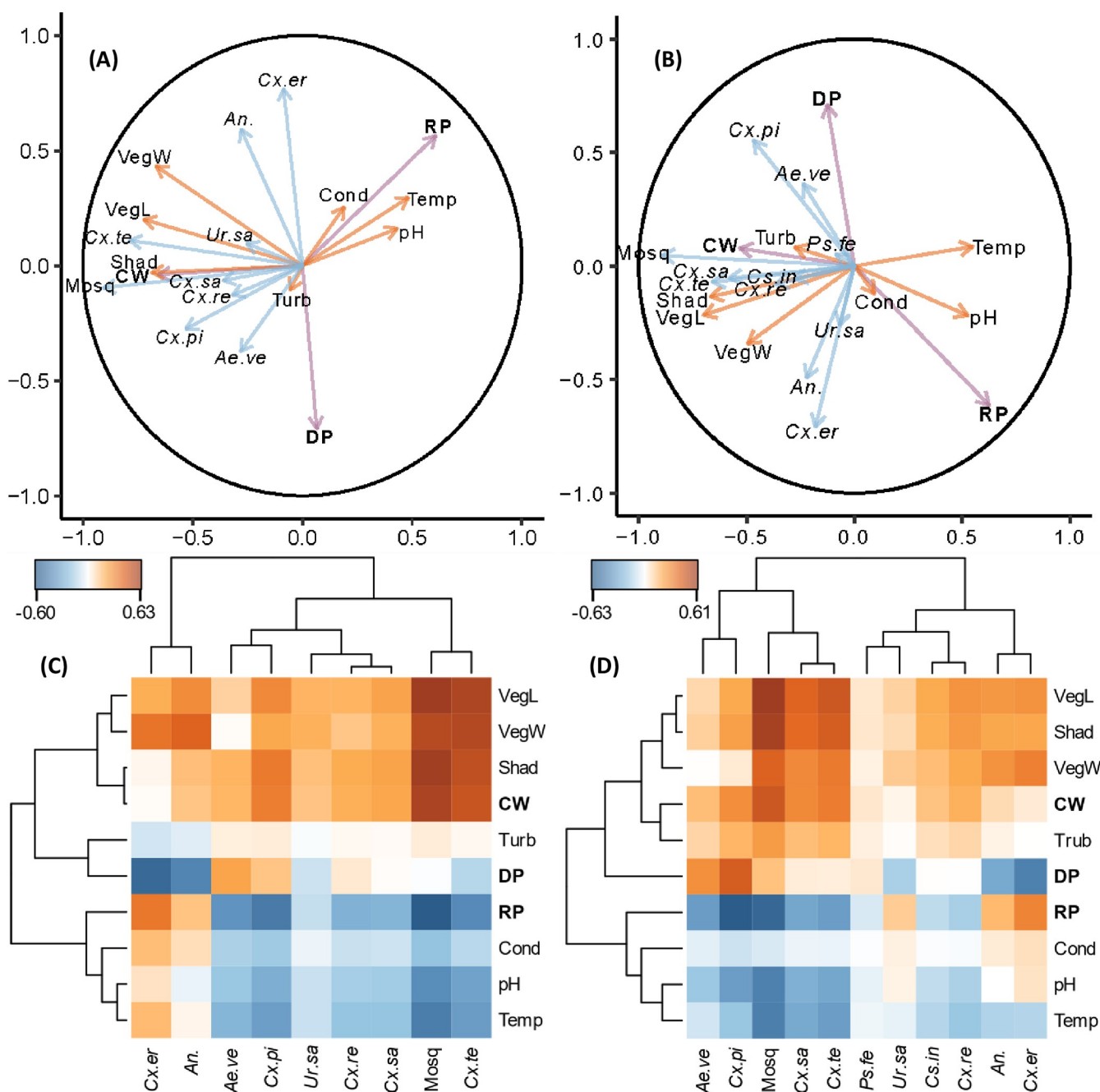

**Fig 2. Partial Least Squares Canonical Analysis (PLSCA) plots representing the correlations between predictor and response variables.** Plots represent data from the (A, C) 2021 season and (B, D) 2022 season. (A, B) Vectors emanating from the origin of the circle to each pair of variables form angles that represent their respective relationships. Blue vectors represent mosquito species or total mosquitoes (response variables), purple vectors represent SCM types (predictor variables), and orange vectors represent environmental parameters (predictor variables). Acute angles represent positively correlated variables with more acute angles indicating a stronger correlation, whereas obtuse angles represent variables that are negatively correlated. The circle in each correlation plot has a radius of 1.0. The length of the vector represents the magnitude of how well the variable can be directly interpreted from the axes chosen for the plot. (C, D) The clustered image maps (CIMs) show the strength and sign of the approximate Pearson correlations between every pair of predictor and response variables. The values used to create these CIMs can be found in S1 Table. The dendrogram associated with each CIM is a representation of the hierarchical clustering used to organize the variables in each CIM. Abbreviations are as follows: *Aedes vexans* (*Ae.ve*), *Anopheles* spp. (*An.*), *Culiseta inornata* (*Cs.in*), *Culex erraticus* (*Cx.er*), *Cx. pipiens* (*Cx.pi*), *Cx. restuans* (*Cx.re*), *Cx. salinarius* (*Cx.sa*), *Cx. territans* (*Cx.te*), *Psorophora ferox* (*Ps.fe*), *Uranotaenia sapphirina* (*Ur.sa*), total mosquitoes (Mosq), constructed wetlands (**CW**), detention ponds (**DP**), retention ponds (**RP**), conductivity (Cond), pH (pH), shade (Shad), temperature (Temp), turbidity (Turb), vegetation on land (VegL), and vegetation within water (VegW).

**Table 7. Mean and standard deviation of alpha diversity indices for each SCM type in 2021.**

| Indices | Retention Ponds | Detention Ponds | Const. Wetlands | p-value |
|---|---|---|---|---|
| Species Richness | 2.63 ± 1.19 | 2.29 ± 1.38 | 4.40 ± 1.35 | < 0.01* |
| Shannon Diversity | 0.63 ± 0.46 | 0.46 ± 0.37 | 0.93 ± 0.33 | 0.09 |
| Shannon Evenness | 0.77 ± 0.26 | 0.71 ± 0.26 | 0.65 ± 0.21 | 0.37 |

Alpha diversity is calculated from total mosquito counts for each site (n = 10 sites/type).

*p-values were determined with Kruskal-Wallis tests testing the main effect of SCM type; significant values are marked with an asterisk.

**Table 8. Mean and standard deviation of alpha diversity indices for each SCM type in 2022.**

| Indices | Retention Ponds | Detention Ponds | Const. Wetlands | p-value |
|---|---|---|---|---|
| Species Richness | 3.38 ± 1.30 | 2.78 ± 1.72 | 4.70 ± 2.11 | 0.11 |
| Shannon Diversity | 0.75 ± 0.38 | 0.44 ± 0.38 | 1.01 ± 0.39 | 0.02* |
| Shannon Evenness | 0.67 ± 0.15 | 0.55 ± 0.26 | 0.67 ± 0.13 | 0.67 |

Alpha diversity is calculated from total mosquito counts for each site (n = 10 sites/type).

*p-values were determined with Kruskal-Wallis tests testing the main effect of SCM type; significant values are marked with an asterisk.

## Effect of SCM type on mosquito diversity

In 2021, species richness of mosquitoes was higher in constructed wetlands than in detention ponds (Table 7; CW vs. DP, p = 0.03), but no significant differences were observed in 2022 (Table 8). In 2022, Shannon diversity of mosquitoes was higher in constructed wetlands than in detention ponds (Table 8; CW vs. DP p = 0.02), but no significant differences were observed in 2021 (Table 7). We did not detect any differences in species evenness among SCM types during either year (Tables 7 and 8).

Patterns of pairwise beta-diversity revealed that SCM types supported similar communities of mosquitoes (Fig 3A and 3B). Although PERMANOVA analyses indicated that community composition of mosquitos differed among SCM types (2021: $r^2$ = 0.161, p = 0.013; 2022: $r^2$ = 0.131, p = 0.026), there were no significant differences using a pairwise comparison with a correction for multiple comparisons. Non-metric multidimensional scaling analyses showed high variability in mosquito community composition among sites, particularly for detention ponds, which likely contributed to these observed patterns (Fig 3).

## Discussion

To improve SCM design and operation, we need a better understanding of how differences in construction and maintenance of SCMs affect their desired and undesired outputs [20]. One important undesired output is the capacity of the SCM to harbor mosquitoes of public health concern. In this study, we compared the abundance and diversity of larval mosquitoes among three different types of SCMs (retention ponds, detention ponds, and constructed wetlands). We demonstrated that retention ponds generally harbored the fewest mosquitoes. We collected the most mosquitoes from constructed wetlands, although these sites tended to be dominated by the innocuous mosquito *Cx. territans*. We also showed that constructed wetlands generally had higher mosquito diversity than the other SCMs. Detention ponds tended to have high mosquito APS and were associated with pest mosquitoes. SCM type only explained a small amount of variation in mosquito APS, indicating that there are other important factors associated with individual sites that affect mosquito abundance. Vegetation, shade,

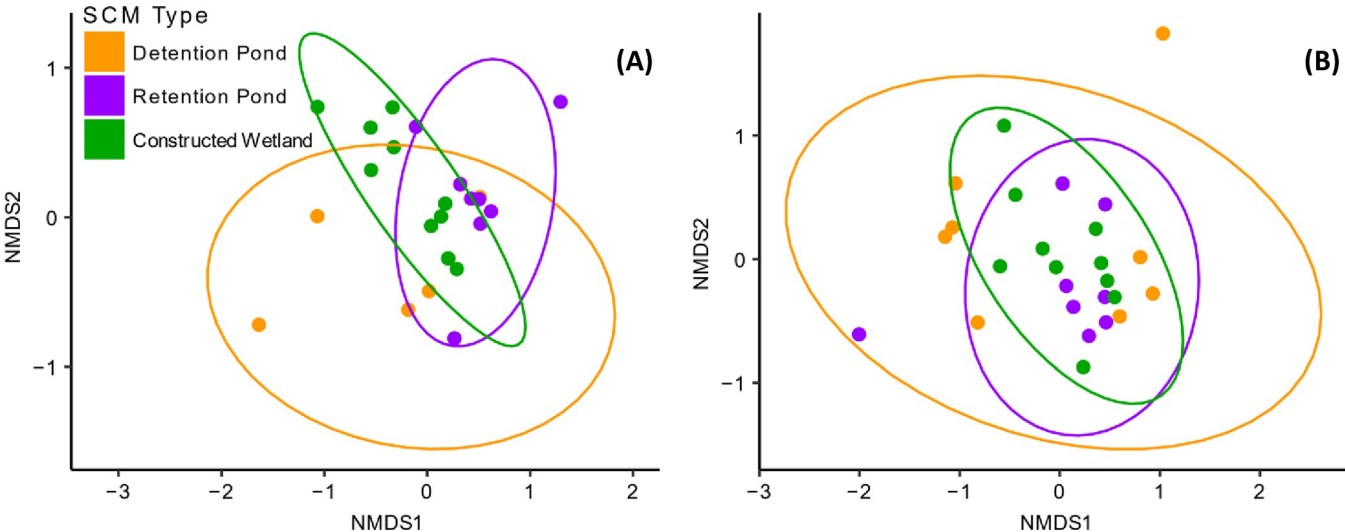

**Fig 3. Non-metric multidimensional scaling analyses using Bray-Curtis dissimilarities of mosquito communities.** Plots represent data from the (A) 2021 season and (B) 2022 season. Each individual point represents a single site, each color represents a different SCM type, and each colored ellipse represents the 95% confidence interval encompassing the population of sites. The NMDS plots were determined to be fair representations of the data (2021: stress = 0.010; 2022: stress = 0.11).

temperature, and pH were correlated with overall mosquito abundance. Because we identified larvae to species, we were also able to show species-specific patterns with SCM types and their associated environmental variables. This is one of the few studies to analyze the effect of SCM type on of mosquito communities in central Ohio [34] or the Midwestern United States [40].

Our study found a total of ten mosquito taxa, including similar species to other SCM studies that have been conducted in the Midwest and Eastern part of the US [9, 28, 34, 40]. While *Cx. territans* was the most commonly found species in our sampling, this species is not known to feed on humans (it typically feed on reptiles and amphibians) or spread pathogens to humans [64, 65]. However, the other four *Culex* species that we collected are all known to feed on both avian hosts and at least occasionally humans [45], making them not only nuisance pests but also potential vectors of enzootic arboviruses such as West Nile virus and eastern equine encephalitic virus. The roles of these four *Culex* species in pathogen transmission have either been well established (e.g., with *Cx. pipiens* [66, 67]) or suspected (e.g., with *Cx. erraticus* [68]). Another common species found in our samples was *Ae. vexans*, a prolific mosquito across the northern hemisphere and beyond. Like many *Aedes* species, it aggressively bites humans and other mammals, making it one of the most important nuisance pests throughout its range. It has also been implicated as a potential vector for several pathogens in the US, Europe, and Asia [66, 69–74]. *Ur. sapphirina* was occasionally found in our samples, although this species is not suspected of being directly involved in human pathogen transmission. The feeding habits of *Ur. sapphirina* have not been well studied, but it is believed to feed primarily on invertebrates [75]. *Anopheles* mosquitoes were commonly found in our sites as well. Although these were not separated by species in our dataset, the two most common species in this region, *An. quadrimaculatus* and *An. punctipennis* [34, 76, 77], are known to bite humans and other mammals and were implicated in the transmission of *Plasmodium* parasites leading to outbreaks of human malaria in the US between the 16th and mid-20th centuries [78], leading to some concern for future outbreaks [3]. They are also well-known vectors of the dog heartworm, *Dirofilaria immitis*, and potential vectors of multiple other human enzootic viruses [79–81]. Finally,

both *Ps. ferox* and *Cs. inornata*, which are potential nuisance pests and pathogen vectors, were found once each in a single constructed wetland site.

Our results showed that constructed wetlands generally harbor a higher abundance and diversity of mosquito larvae than retention ponds and detention ponds. The propensity of wetlands to generate mosquitoes is a common concern [31, 37] and an important reason why the public is sometimes resistant to the installation of wetlands. However, studies have also demonstrated that increasing the number of wetlands and the areas of standing water associated with them is correlated with a decreasing incidence of West Nile virus in mosquitoes [82–85]. Our results support the notion that wetlands produce more mosquitoes. However, these constructed wetlands were most strongly associated with harmless mosquito species (*Cx. territans* and *Ur. sapphirina*). On the other hand, detention ponds supported a lower diversity of mosquitoes and were more strongly associated with *Ae. vexans* and *Cx. pipiens*, species that are known to bite humans more aggressively and potentially vector human pathogens. Detention ponds are characterized by periods of both inundation and drought, an environmental pattern that is known to be conducive to the life history patterns of *Ae. vexans* [86] and potentially *Cx. pipiens* [87]. Additionally, other studies have shown that detention ponds can harbor similar or higher numbers of mosquitoes than constructed wetlands [28, 35]. In some cases, correlations between individual mosquito species and SCM types were stronger in one year of the study than the other. For example, *Cx. pipiens* only showed a strong positive correlation with detention ponds in 2022, and *Anopheles* spp. only showed a strong negative correlation with detention ponds in 2021). Although these differences may be due to natural fluctuations in the populations of each mosquito species on a broad scale, they may also be related to differences in weather patterns. For example, if *Cx. pipiens* thrive better under drought/inundation conditions [87], then they may have a stronger correlation with detention ponds on years where this pattern occurs more frequently. Further analysis of weather patterns and water level changes in SCMs may help elucidate these relationships.

Overall, retention ponds harbored the fewest mosquitoes compared to the other SCM types. The only taxa that were positively correlated with retention ponds were *Cx. erraticus* and *Anopheles* spp., although these taxa were also associated with constructed wetlands. The bathymetric profile of retention ponds (generally deeper depths with steep side slopes) [29] may reduce the habitat available for mosquitoes. Previous literature suggests that deeper water and steeper side slopes are not suitable for mosquitoes [28, 30]. Other types of SCMs, particularly constructed wetlands, are likely to be shallower or feature more shallow areas of water [24]. In our study, we chose to only target depths of 5 cm and 15 cm to provide consistency in our experimental design and because shallower depths are more suitable for mosquitoes [29]. If this is the case, then we may be underrepresenting the difference in the number of mosquitoes harbored in constructed wetlands versus retention ponds because constructed wetlands feature a greater percentage of shallow water.

Shallower water in constructed wetlands provides suitable areas for vegetation to grow. One of the main differences between constructed wetlands and the other SCM types is that constructed wetlands are typically planted with a high diversity of native hydrophytic vegetation. Constructed wetlands sampled in this study were more likely to have more shade, vegetation within water, and vegetation on land, and these factors were all positively correlated with mosquito abundance. The presence of vegetation in constructed wetlands may be one of the main factors that is more attractive to ovipositing adult mosquitoes and supportive of larval development. Although detention and retention ponds are not typically planted with vegetation, invasive plants like *Typha* spp. [88] and *Phragmites* spp. [89] commonly establish in SCMs of these types. Constructed wetlands can also get overrun with these species over time, as they replace the hydrophytic vegetation that is intentionally planted during initial

construction [88]. Collins and Resh provide a table of common wetland plants and their propensity towards supporting mosquito development based on ecological parameters of the plant and predict that *Typha* spp. and *Phragmites* spp. are among the most supportive plants towards mosquito development [39]. However, relationships between these invasive plants and mosquito larvae are inconsistent. Some research has found a positive relationship [90, 91] and others show a negative [34, 92, 93] or non-significant relationship [40, 41, 94, 95]. Management of vegetation in SCMs can be expensive and difficult to perform in a way that minimizes mosquito production [96]. Therefore, many SCMs become overrun with undesired vegetation, leading to increased habitat suitable for mosquitoes. To avoid this, more research should be done on identifying parameters of SCMs that will make them refractory to the establishment of these less-desired plants, lowering the overall cost and effort to maintain the SCMs in a way that minimizes mosquito production.

Our study showed a strong positive correlation between shade and vegetation as well as total mosquitoes, indicating the possibility that one of the main benefits of vegetation to mosquitoes is the provision of shade, as has been suggested in the past [97]. Vegetation and the resultant shade it provides may lead to lower water temperatures which are associated with higher mosquito abundance in our dataset. Additionally, vegetation may increase the abundance of adult mosquito hosts (e.g., birds, small mammals, amphibians, reptiles), indirectly increasing the number of mosquitoes near a site [95]. Another possibility is that the vegetation in or near the water may create increased food availability for larval mosquitoes, which are filter feeders, in the form of detritus or bacteria that feed upon detritus [41, 94, 98].

One caveat of our study is that we used a larval dipper to extract larvae that typically breathe air at the surface of the ponds, similar to other SCM studies [9, 28, 34, 40, 42]. This method may not be well suited to collecting larvae in the tribe Mansoniini which embed their siphon into plants to breathe and therefore spend limited time near the surface [99, 100]. For example, *Coquillettidia perturbans* (Walker) is commonly found around central Ohio [76, 77, 101] and is a potential vector of human pathogens [80, 102]. It is possible that we did not find this species in our study, not because it was not present, but instead because we did not use specific techniques to sample this species [100]. More targeted collection methods may be needed to address some of the more unusual mosquito species.

Another important caveat to consider is the life stages of the mosquitoes that we sampled. We recorded all instar stages as well as pupae, but these numbers may not represent mosquitoes that mature fully, particularly if they are disproportionately killed by predators [40], plant toxins [103], or other factors that cause sites to be attractive for oviposition but deleterious to maturation and emergence. One way to overcome this issue is to collect adult mosquitoes around SCM sites using carbon dioxide or gravid traps (such as in [33, 36, 104]) or directly from SCMs using emergence traps. With the exception of emergence traps, adult trapping presents its own problems as it may be difficult to determine if the adults collected actually emerged from the specified SCM and because these traps may introduce bias based on the species that are most attracted to them.

As the number of SCMs grows over time, it is becoming increasingly important to recognize the potential of SCMs to produce mosquitoes and find ways of minimizing their potential as a public health hazard. Our study supports the notion that not all SCMs share the same potential to produce mosquitoes. Our findings indicate that constructed wetlands produce the greatest number of mosquitoes, however, a large proportion of these mosquitoes may be innocuous to humans. On the other hand, we collected a lower total number of mosquitoes as well as a lower diversity of mosquitoes from detention ponds, but these sites tended to harbor a higher number of potential vectors such as *Ae. vexans* and *Cx. pipiens*. In general, we observed that retention ponds harbored the fewest mosquitoes. Because of these results, it may

be beneficial to incorporate aspects of retention ponds in SCM designs. For example, constructing wetlands with steeper slopes and larger deep-water zones may reduce mosquito production, although these parameters should be balanced with the concern for public safety as this may increase the risk for drownings [96]. Additionally, ensuring that constructed wetlands and detention ponds do not hold water in inundation zones for long enough time to allow larvae to develop fully (e.g., by changing discharge rates) will also reduce mosquito production. Our study also showed that vegetation and environmental parameters associated with vegetation (shade, water temperature) were among the most important factors that determined mosquito abundance. Therefore, more research on best practices for vegetation management is needed. Additionally, as previous studies have shown that different species of plants can be more or less favorable to mosquitoes, careful thought should be given to choosing which species to plant in SCMs like constructed wetlands. With a better understanding of these factors, coupled with an appreciation of the varied impact of certain interventions on different mosquito species, we can help improve the guidelines for how to best construct and maintain SCMs in ways that minimize the harmful impacts of mosquitoes.

## Supporting information

**S1 File. Raw data excel spreadsheet.** Each row represents an individual sample, and each column represents each recorded variable. The second tab provides a key for the variables, their possible values, and the units. Blank values represent no data collected (e.g., because a site was dry during that sampling period).
(XLSX)

**S2 File. R markdown file with R code and output for statistical tests.**
(HTML)

**S3 File. Mosquito identification confirmation using COI barcoding.** The COI sequences and results of the GenBank BLAST query are recorded for each individual mosquito. The BLAST search result, percent identity, and query cover are recorded for the top result; two species are listed if both share the same similarity to the queried sequence.
(XLSX)

**S1 Table. Similarity matrices between mosquito taxa and environmental variables.** The values of the matrices are estimates of Pearson correlation coefficients between each pair of variables. The values are generated using PLSCA results.
(XLSX)

## Acknowledgments

We wish to thank the private property managers and government offices for allowing our use of their properties for this study. We would also like to acknowledge Dr. Mary Gardiner for providing advice on our sampling design as well as providing materials for this study. Finally, we would like to acknowledge Cade Capper, Rachael Kaiser, Spencer Blankenship, and Camden Dezse for their assistance with collecting samples.

## Author Contributions

**Conceptualization:** James Radl, R. Andrew Tirpak, Ryan J. Winston, Sarah M. Short.

**Data curation:** James Radl.

**Formal analysis:** James Radl, Luis Martínez Villegas, Kayla I. Perry, Sarah M. Short.

**Funding acquisition:** R. Andrew Tirpak, Ryan J. Winston, Sarah M. Short.

**Investigation:** James Radl, Luis Martínez Villegas, Joseph S. Smith, R. Andrew Tirpak, Deirdre Wetmore, Elena Tunis, Jack Smithberger, Henry Schuellerman, Dom Magistrado, Sarah M. Short.

**Methodology:** James Radl, Joseph S. Smith, R. Andrew Tirpak, Deirdre Wetmore, Ryan J. Winston, Sarah M. Short.

**Project administration:** James Radl, Joseph S. Smith, Ryan J. Winston, Sarah M. Short.

**Supervision:** James Radl, Joseph S. Smith, Ryan J. Winston, Sarah M. Short.

**Validation:** James Radl.

**Visualization:** James Radl, Luis Martínez Villegas, Kayla I. Perry, Sarah M. Short.

**Writing – original draft:** James Radl.

**Writing – review & editing:** James Radl, Luis Martínez Villegas, Joseph S. Smith, R. Andrew Tirpak, Kayla I. Perry, Dom Magistrado, Ryan J. Winston, Sarah M. Short.

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
