## [Decision Letter · Decision Letter 0]

17 Apr 2024

PONE-D-24-07755

Mosquito abundance and diversity in central Ohio vary among stormwater wetlands, retention ponds, and detention ponds as well as associated environmental parameters

PLOS ONE

Dear Dr. Short,

Thank you for submitting your manuscript to PLOS ONE. After careful consideration, we feel that it has merit but does not fully meet PLOS ONE’s publication criteria as it currently stands. Therefore, we invite you to submit a revised version of the manuscript that addresses the points raised during the review process.

Please **fully address** the issues raised by the reviewers. Although it is a minor revision, some detailed and significant changes need to be done before acceptance. As suggested by both reviewers, please think about revising the title of the manuscript.

We look forward to receiving your revised manuscript.

Kind regards,

Carina Zittra, Ph.D., Mag.rer.nat.

Academic Editor

PLOS ONE

Journal Requirements:

Reviewers' comments:

Reviewer's Responses to Questions

**Comments to the Author**

1. Is the manuscript technically sound, and do the data support the conclusions?

Reviewer #1: Yes

Reviewer #2: Yes

2. Has the statistical analysis been performed appropriately and rigorously? 

Reviewer #1: Yes

Reviewer #2: Yes

3. Have the authors made all data underlying the findings in their manuscript fully available?

Reviewer #1: Yes

Reviewer #2: Yes

4. Is the manuscript presented in an intelligible fashion and written in standard English?

Reviewer #1: Yes

Reviewer #2: Yes

5. Review Comments to the Author

Reviewer #1: Radl et al. describe variations in mosquito abundances in different habitats (stormwater control measures) in central Ohio, USA.

The manuscript is of good quality, relevance and interest. Congratulations.

Some modifications are recommended.

Barcoding of the mt COI was done. Please upload your sequences to GenBank and provide IDs.

Please include USA in the title.

Change Cx. pipiens to Cx. pipiens s.l. or Cx. pipiens complex

L53: only pathogens can be spread - not diseases

L 62: Point afer (6)

L 147 and entire manuscript: Space after SI Unit (e.g. 50 cm)

Line 176, 411, 412: Diseases cannot be transmitted

L 326: There are differences at same sampling locations in two years - different species - please discuss

Line 427: Malaria is the disease caused by Plasmodium spp.

Line 499: Point after spp.

Reviewer #2: Overall, a nice study that adds to our understanding of mosquito production from stormwater management structures in urban environments

A few comments that I think should be considered:

- Approximately 25 years ago, engineered structures designed to manage stormwater flows, recharge groundwater, capture trash, sediment, or other target pollutants began to be referred to collectively as “Best Management Practices”. Unfortunately, other actions related to stormwater and stormwater runoff also fell under the “BMP” umbrella, from policies to street sweeping to vegetation management. This was unnecessarily confusing, and fortunately the stormwater community has moved away from using this generic term when referring to engineered stormwater structures and replaced it with more specific names that define the category such as “trash capture devices”, “ground water recharge basins”, “constructed stormwater treatment wetlands”, etc.

With that said, I feel that your decision to use “Stormwater Control Measures – SCM” is similarly vague, like BMP, since “measures” here again could be any number of things including policy, actions, or structures. I strongly recommend you consider using a name that better indicates that these are engineered structures, e.g., Stormwater Management Structures; Stormwater Control Structures, etc.

- The design and proper operation of any stormwater management device / structure falls apart without adequate maintenance. However, maintenance schedules and physical maintenance requirements are not universal and usually have to be considered at a minimum based on the category of a structure (e.g., detention basins), but more often on a site per site basis depending on the size of the structure, the amount of natural and urban generated (non-stormwater) runoff, trash and sediment loads entering, etc.

You tease us with the importance of maintenance in 3 areas of the manuscript: the introduction (lines 123-25), in the discussion opening paragraph (lines 393-394), and in the concluding paragraph of the paper (line 521). I would like to see a minimum of some thoughts on how the results of this study might inform future maintenance schedules, activities, and objectives (at least for the local study area). We know from experience that sediment loading can have severe impacts on operation of the structure and mosquito production. We know that steep sides are often not utilized by engineers due to public safety concerns. We know that design vegetation almost never holds and is eventually replaced by monocultures of Typha. There are so many issues related to design and maintenance. Please explore some ideas on how mosquito production could be better controlled / minimized for both nuisance biting and disease management. Why is maintenance important? How do requirements differ among the 3 categories of structures in your study?

- A few editing and suggested odds and ends:

o I think you should consider revising the title of the paper. In particular, the “as well as associated environmental parameters” feels like an afterthought.

o Line 41. Replace determine with elucidate

o Line 57. YF and malaria are no longer endemic in the USA. Revise to “Although diseases such as YF and malaria….”

o Line 60. Revise sentence "Mosquitoes in the US can transmit lethal….”

o Line 77. Consider including “available land / real estate” to this list

o Lines 121-123. This sentence belongs in the Discussion

o Line 130. “categories” is probably a better choice over “types”

o Line 176. Remember that mosquitoes do not transmit “diseases”, they transmit pathogens.

o Line 395. I think “harmful” should not be used when referring to certain species of mosquitoes, but rather “mosquitoes of public health significance or concern”

o Line 511. As I mentioned above, we would love to see all basins built with steep sides to limit vegetation growth / shallow areas, and therefore reduce available larval habitat, but civil engineers often have to consider public safety (i.e. drownings) when designing and building stormwater control structures, which is why we don’t often see the steep sides that we wish for.

o Line 516. I suggest you look at Bill Walton’s paper (UCANR Publication 8117; Managing mosquitoes in surface-flow contructed treatment wetlands). https://anrcatalog.ucanr.edu/Details.aspx?itemNo=8117

o Line 518. Also mentioned above, I have yet to see a stormwater management structure / constructed treatment wetland that did not soon become a monoculture of cattails and /or bullrush. Although designs with fancy planting proposals look great on paper and are easy to sell, the maintenance required to keep the plant community as designed is beyond the scope and expertise of most municipalities. It's simply too labor intensive and specific in the majority of cases.

6. PLOS authors have the option to publish the peer review history of their article (what does this mean?). If published, this will include your full peer review and any attached files.

Reviewer #1: **Yes: **Hans-Peter Fuehrer

Reviewer #2: No

---

## [Author Response · Author response to Decision Letter 0]

20 May 2024

See Response to Reviewers document

---

## [Editor Report · Decision Letter 1]

30 May 2024

Mosquito abundance and diversity in central Ohio, USA vary among stormwater wetlands, retention ponds, and detention ponds and their associated environmental parameters

PONE-D-24-07755R1

Dear Dr. Short,

We’re pleased to inform you that your manuscript has been judged scientifically suitable for publication and will be formally accepted for publication once it meets all outstanding technical requirements.

Kind regards,

Carina Zittra, Ph.D., Mag.rer.nat.

Academic Editor

PLOS ONE
---

## [Editor Report · Acceptance letter]

14 Jun 2024

PONE-D-24-07755R1 

PLOS ONE

Dear Dr. Short, 

I'm pleased to inform you that your manuscript has been deemed suitable for publication in PLOS ONE. Congratulations! Your manuscript is now being handed over to our production team.

Kind regards, 

on behalf of

Dr. Carina Zittra 

Academic Editor

PLOS ONE